# *Drosophila* Caliban preserves intestinal homeostasis and lifespan through regulating mitochondrial dynamics and redox state in enterocytes

Zhaoxia Dai[1]☯, Dong Li[1,2]☯*, Xiao Du[1,2], Ying Ge[1], Deborah A. Hursh[3], Xiaolin Bi[2,4]*

1 The Second Affiliated Hospital, Institute of Cancer Stem Cell, Dalian Medical University, Dalian, China, 2 School of Medicine, Nantong University, Nantong, China, 3 Division of Cellular and Gene Therapies, Office of Tissues and Advanced Therapies, Center for Biologics Evaluation and Research, U.S. Food and Drug Administration, Silver Spring, Maryland, United States of America, 4 College of Basic Medical Sciences, Dalian Medical University, Dalian, China

☯ These authors contributed equally to this work.
* lidong@dmu.edu.cn (DL); bixl@ntu.edu.cn (XB)

**Data Availability Statement:** The raw microarray data was deposited at https://www.ebi.ac.uk/arrayexpress/experiments/E-MEXP-1817/?query=

## Abstract

Precise regulation of stem cell activity is crucial for tissue homeostasis. In *Drosophila*, intestinal stem cells (ISCs) maintain the midgut epithelium and respond to oxidative challenges. However, the connection between intestinal homeostasis and redox signaling remains obscure. Here we find that Caliban (Clbn) functions as a regulator of mitochondrial dynamics in enterocytes (ECs) and is required for intestinal homeostasis. The *clbn* knock-out flies have a shortened lifespan and lose the intestinal homeostasis. Clbn is highly expressed and localizes to the outer membrane of mitochondria in ECs. Mechanically, Clbn mediates mitochondrial dynamics in ECs and removal of *clbn* leads to mitochondrial fragmentation, accumulation of reactive oxygen species, ECs damage, activation of JNK and JAK-STAT signaling pathways. Moreover, multiple mitochondria-related genes are differentially expressed between wild-type and *clbn* mutated flies by a whole-genome transcriptional profiling. Furthermore, loss of *clbn* promotes tumor growth in gut generated by activated Ras in intestinal progenitor cells. Our findings reveal an EC-specific function of Clbn in regulating mitochondrial dynamics, and provide new insight into the functional link among mitochondrial redox modulation, tissue homeostasis and longevity.

## Author summary

Self-renewal and differentiation of somatic stem cells are critical for tissue homeostasis. In *Drosophila*, intestinal homeostasis is maintained by tightly controlled proliferation and differentiation of intestinal stem cells (ISCs). In this study, we demonstrate that maintenance of mitochondrial dynamics and redox balance in enterocytes (ECs) by Caliban (Clbn) is critical for ISCs proliferation and intestinal homeostasis. We show that *clbn* mutant flies have shortened lifespan, involving disruption of intestinal homeostasis. We

Mortin. The main analysis script can be found at https://github.com/ying-ge/clbn.

**Funding:** This work was supported by grants from National Natural Science Foundation of China to XB (Grant No. 31771437, 31970605) and DL (Grant No. 31501165). The funders had no role in study design, data collection and analysis, decision to publish, or preparation of the manuscript.

find that Clbn is highly expressed and localized to the outer membrane of the mitochondria in enterocytes. Clbn is important for mitochondrial dynamics, as loss of *clbn* in enterocytes results in mitochondrial fragmentation. Mitochondrial defects cause accumulation of reactive oxygen species (ROS) production and cellular damage, which in turn leads to activation of oxidative stress and promotion of ISCs over-proliferation. We further show that depletion of *clbn* promotes tumor growth in gut generated by activated Ras in intestinal progenitor cells. Our results establish Clbn as a modulator of mitochondrial dynamics in enterocytes, and highlight the importance of mitochondrial dynamics in the regulation of somatic stem cells activity and tissue homeostasis.

## Introduction

Tissue homeostasis needs to be properly maintained to preserve the organismal fitness, especially in tissues with high turnover rate, such as the intestinal epithelium. The digestive tract is constantly challenged by chemicals, toxins, and pathogens in ingested food, and needs to be regenerated continuously. The intestinal stem cells (ISCs) exhibit the ability to self-renew to maintain the stem cell population, generate daughter cells of all mature cell types, and replace damaged and aged cells during normal epithelial regeneration as well as in response to cellular stress and injury. These functions are essential for intestinal homeostasis. Deregulation of ISCs proliferation or differentiation can lead to developmental abnormality and chronic diseases [1].

The *Drosophila* intestinal system shares anatomical and functional similarities with the mammalian digestive system, and is a facile model for dissecting the signaling mechanisms that control stem cell self-renewal and differentiation [2]. The *Drosophila* ISCs originate from adult midgut precursors (AMPs), the endodermal progenitors, during the larval stage [3], and reside adjacent to the basement membrane [4, 5]. ISCs self-renewal and differentiation are controlled mainly by differentially regulated Notch pathway activity. The daughter cells with high Notch activity become transient, differentiating precursor cells called enteroblasts (EBs), which terminally differentiate into enterocytes (ECs). The daughter cells with low Notch activity retain ISCs identity and are primed to differentiate into pre-enteroendocrine (pre-EE) cells and further into enteroendocrine (EE) cells when the neuronal transcription factor, Prospero is highly expressed [4–10]. Multiple evolutionarily conserved signaling pathways are required to maintain normal ISCs activities, such as Notch, EGFR, Wnt, JAK-STAT, JNK, BMP, Hippo, Slit/Robo etc [7, 11–17].

Mitochondria are highly dynamic organelles that continually fuse and divide. The morphology of mitochondria is well balanced between fusion and fission reactions and linked with ROS production [18, 19]. Defective mitochondrial fusion and fission are often associated with aging, metabolic disease, and cancer [20]. The regenerative potential of stem cells is closely correlated with the level of intracellular reactive oxygen species (ROS). Normal physiological levels of ROS are essential for stem cells functions and fate decision [21,22]. Mitochondria are the principal source of cellular ROS (mtROS), and a regulated physiological level of mtROS is important to maintain stem cells self-renewal and differentiation. Recent studies suggest that genes regulating mitochondrial fusion-fission dynamics play important roles for the maintenance of stem cells functions, fate decisions and aging processes [23–26].

Similar to mammals, *Drosophila* intestinal stem cells also have increased proliferation in response to oxidative stress. Commensal bacteria in the *Drosophila* gut can induce NADPH oxidase 1 (NOX1)-dependent endogenous ROS generation and proliferation of intestine stem cells [27], and enteropathogenic bacteria *ECC-15* infection induces oxidative burst, activates

JNK and JAK-STAT signaling pathways and increases ISCs proliferation [28]. Increased level of unfolded protein response of the endoplasmic reticulum (UPR$^{ER}$) in tissues can activate the JNK pathway by producing ROS and cause hyper-proliferation of ISCs [29]. The master regulator of cellular redox state, Nrf2, maintains intestinal homeostasis through controlling ISCs proliferation [30]. Mitochondrial dynamics also regulate differentiation of ISCs in *Drosophila*; knock-down of *opa1* or *mitofusin* (*mfn*) to inhibit mitochondrial fusion causes differentiation failure [31]. Although the roles of mitochondria in stem cell self-renewal and differentiation are becoming more evident, many questions are still open. Further work needs to be done to identify genes related to mitochondria dynamics and metabolism, and to decipher their functions in the maintenance of stem cell homeostasis.

We have previously shown that *Drosophila* Caliban (Clbn) works as a nuclear exporting mediator for Prospero in S2 cells and a tumor suppressor in human NSCLC cells [32]. We generated *clbn* gene knock-out fly (a deletion covering the entire *clbn* protein-coding region) using homologous recombination, and demonstrated that Clbn regulates DNA damage induced p53-dependent and–independent cell apoptosis, and S phase cell cycle checkpoint by antagonizing E2F1 activity [33–34]. A recent study demonstrated that Clbn can bind to fly Ala- and Thr-tRNAs, and loss of *clbn* rescued the mitochondrial morphology and DA neuron loss phenotypes in muscle of *PINK1* mutant, although *clbn* mutant showed normal phenotype [35]. In the present study, we carried out series of genetic analyses, and provided the direct evidence that *Drosophila* Clbn is required for the maintenance of adult intestinal homeostasis and lifespan by specifically regulating mitochondrial dynamics and redox balance in enterocytes. Flies loss of *clbn* have shortened lifespan. Clbn resides in mitochondrial outer membrane in enterocytes. Depletion of *clbn* in enterocytes results in mitochondrial defects and accumulation of ROS production and cellular damage, which in turn leads to activation of oxidative stress and promotion of ISCs hyper-proliferation. Thus, our results demonstrate the novel function of Clbn in regulating mitochondrial dynamics and intestinal homeostasis and further highlight the importance of mitochondrial dynamics control in regulating somatic stem cells proliferation and tissue homeostasis.

## Results

### Flies loss of *clbn* have shortened lifespan and dysfunctional intestinal barrier

Flies loss of *clbn* gene are viable, show no external morphological defects and small developmental delays under physiological conditions [33]. However, *clbn* knock-out flies exhibited a reduced lifespan compared with control flies, and most of *clbn* mutated flies died at eight weeks after birth (Fig 1A). Consistently, the *clbn* knockdown flies had reduced lifespan, and restoration of *clbn* expression can fully rescue the lifespan of *clbn* knock-out flies (Fig 1A). As defects in the maintenance of intestinal homeostasis could result in shortened lifespan in flies [36], we assessed whether intestinal morphology and integrity were damaged in *clbn* mutant. We did not observe a morphology change in guts from *clbn* mutant adults at 15-day-old. We next examined integrity of the intestinal barrier by feeding flies with non-absorbable blue food dye. In wild-type flies, blue food dye was restricted to the pro-boscis and in digestive tract, whereas blue dye spread to other body parts in *clbn* mutant, indicating the intestinal barrier was dysfunctional in *clbn* mutant (Fig 1B–1E).

### Loss of *clbn* causes over-proliferation of intestinal stem cells

To explore the cause of defective intestinal barrier in *clbn* mutant, we defined the number of ISCs, EBs, EEs and ECs in the fly midgut. To distinguish the different cell types in intestinal

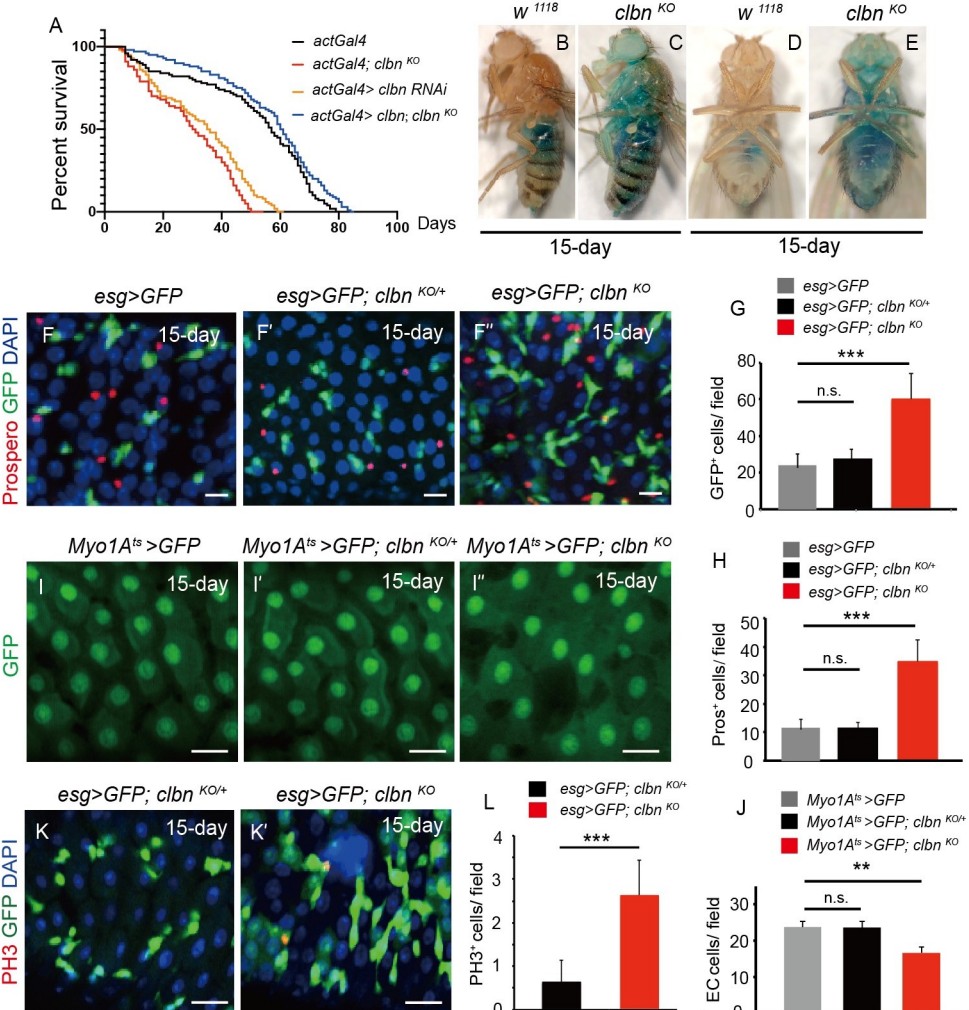

**Fig 1. Loss of *clbn* leads to shortened lifespan and adult intestinal defect.** (A) Survival curves of indicated genotypes. *Clbn* knock-out or knock-down flies exhibited shortened lifespan compared with control flies. n > 100. (B, D) 15-day-old wild-type female flies were fed with food containing a nonabsorbed food dye (FD&C blue dye #1). (C, E) 15-day-old *clbn* $^{KO}$ female flies consumed the same food dye exhibited "smurf" phenotype. (F-F") The posterior midguts of 15-day-old control (*esg* >*GFP*), *clbn* $^{KO/+}$ and *clbn* $^{KO}$ flies stained with anti-Prospero antibody (red) and DAPI (blue). (G, H) Quantification of the number of *esg-GFP*$^{+}$ cells (G) or Prospero$^{+}$ (H) cells in control (n = 12), *clbn* $^{KO/+}$ (n = 13) and *clbn* $^{KO}$ flies (n = 10). (I-I") The posterior midguts of control (*Myo1A* $^{ts}$ >*GFP*), *clbn* $^{KO/+}$ and *clbn* $^{KO}$ flies aged at 29°C for 15 days. (J) Quantification of the number of EC cells in control (n = 15), *clbn* $^{KO/+}$ (n = 16) and *clbn* $^{KO}$ flies (n = 13). (K-K') The posterior midguts of 15-day-old *clbn* $^{KO/+}$ and *clbn* $^{KO}$ flies stained with anti-phosphorylated histone H3-PH3 antibody and DAPI. (L) Quantification of the number of PH3$^{+}$ cells in *clbn* $^{KO/+}$ (n = 8) and *clbn* $^{KO}$ flies (n = 10). The data shown are means ± SEM, and *P* value was noted as follows: $^{**}P < 0.01$, $^{***}P < 0.001$. Scale bars: 20 um.

epithelium in female adult flies, we used *esgGal4>GFP* to label progenitor cells (ISCs and EBs), *Myo1A>GFP* to label ECs, and anti-Prospero antibody staining to label EEs. In young flies (5-day-old), loss of *clbn* did not significantly affect the number of ISCs, EBs and EEs (S1 Fig). However, *clbn* mutant at day 15 or 30 after birth had significantly increased number of ISCs and EBs when compared with that of control flies (S1 Fig), while the number of progenitor cells was comparable between *clbn* heterozygous flies and control flies, suggesting increased proliferation of progenitor cells and rapid epithelial turnover in *clbn* mutant (Fig 1F–1F" and 1G). We further used the ISC marker *delta-LacZ* and the EB marker *Su(H)GBE>GFP* to

distinguish ISCs from EBs, and observed a marked increase of both ISCs and EBs in the *clbn* mutant (S2 Fig). Furthermore, the number of EE cells (Pros+) was significantly increased in the *clbn* mutant (Fig 1F", 1H and S1 Fig), however the number of EC cells was reduced (Fig 1I–1I" and 1J). Immunostaining with anti-phospho-histone H3 (PH3) antibody revealed a significant increase of PH3$^+$ cells in *clbn* mutant, and most of PH3$^+$ cells were *esgGal4>GFP$^+$* (Fig 1K′, 1L and S1 Fig), indicating the over-proliferation of intestine stem cells in *clbn* mutant.

To address the possibility that the changed number of ISCs, EBs, EEs and ECs in *clbn* mutant resulted from a block to differentiation, we performed the mosaic analysis with a repressible cell marker (MARCM), and detected the presence of EEs with anti-Prospero antibody and ECs with anti-Pdm1 antibody in *clbn* mutant clones, respectively. The EEs and ECs were presented within *clbn* mutant clones without significant changes in cell numbers (S3A– S3F Fig), indicating that *clbn* is not required for epithelial cell differentiation in the midgut. Moreover, the cell numbers in MARCM clones were comparable between wild-type and *clbn* mutant, suggesting that Clbn might function non-autonomously in regulating ISC proliferation (S3H Fig). Together, these findings indicated that *clbn* is essential for the control of ISCs proliferation and intestinal homeostasis, but dispensable for cell differentiation.

## Clbn resides in enterocytes to modulate intestinal homeostasis

Next we detected the cellular localization of Clbn in the fly intestinal epithelium using anti-Clbn antibody staining. Clbn protein was highly expressed in *Myo1A>GFP$^+$* ECs, and most of Clbn protein was cytoplasmic (Fig 2A and 2A'). However, the *esg>GFP$^+$* progenitor cells or Pros$^+$ EEs had little or no expression of Clbn protein (Fig 2B–2C'). The expression of *clbn* was confirmed by qPCR on sorted ISC/EB, EE and EC cells using fluorescence-activated cell sorting (FACS) (Fig 2D). To further explore the site that Clbn acts in the midgut, we used a panel of Gal4 drivers to drive knock-down of *clbn* in the different cell types in midgut. The number of progenitor cells and EE cells were significantly increased when *clbn* was knocked-down in the ECs under *Myo1A-Gal4* (S4A, S4B, S4F and S4G Fig), while knock-down of *clbn* in the ISC and EB cells under *esg-Gal4*, or knock-down of *clbn* in the EE cells under *pros-Gal4*, had no significant effect on the number of ISC, EB and EE cells (S4C, S4D, S4F and S4G Fig). Moreover, knock-down of *clbn* in the smooth muscle surrounding the gut with *mef2-Gal4* had no effect on the number of ISC, EB and EE cells (S4E–S4G Fig), suggesting that Clbn performs its functions specifically in ECs to regulate intestinal homeostasis.

To further confirm that Clbn works in the ECs to regulate intestinal homeostasis, we performed rescue experiments. Forced expression of *clbn* in ECs using *Myo1A-Gal4* completely rescued the abnormal increased number of progenitor cells and EE cells in the *clbn* mutant, while expression of *clbn* in progenitor cells using *esg-Gal4* partially rescued dysplasia in the *clbn* mutant (Fig 2E–2J), indicating that Clbn functions mainly in ECs to regulate intestinal homeostasis. Furthermore, the shortened lifespan of *clbn* mutant can be largely rescued by expressing *clbn* in ECs (Fig 2K), while no rescue effect was observed when expressing *clbn* in ISCs and EBs (Fig 2L).

## Clbn regulates mitochondrial dynamics in ECs

The cytoplasmic localization of Clbn in ECs shows a "perinuclear" pattern, and Clbn protein is highly enriched around nucleus (Fig 2A and 2A'), which suggested that Clbn might be localized to mitochondria. To test this hypothesis, we used a mito-GFP reporter driven by *Myo1A-Gal4* to label the mitochondria in the ECs. The Clbn protein predominantly colocalized with the mito-GFP in the ECs (Fig 3A–3C‴). Furthermore, by performing cell fractionation

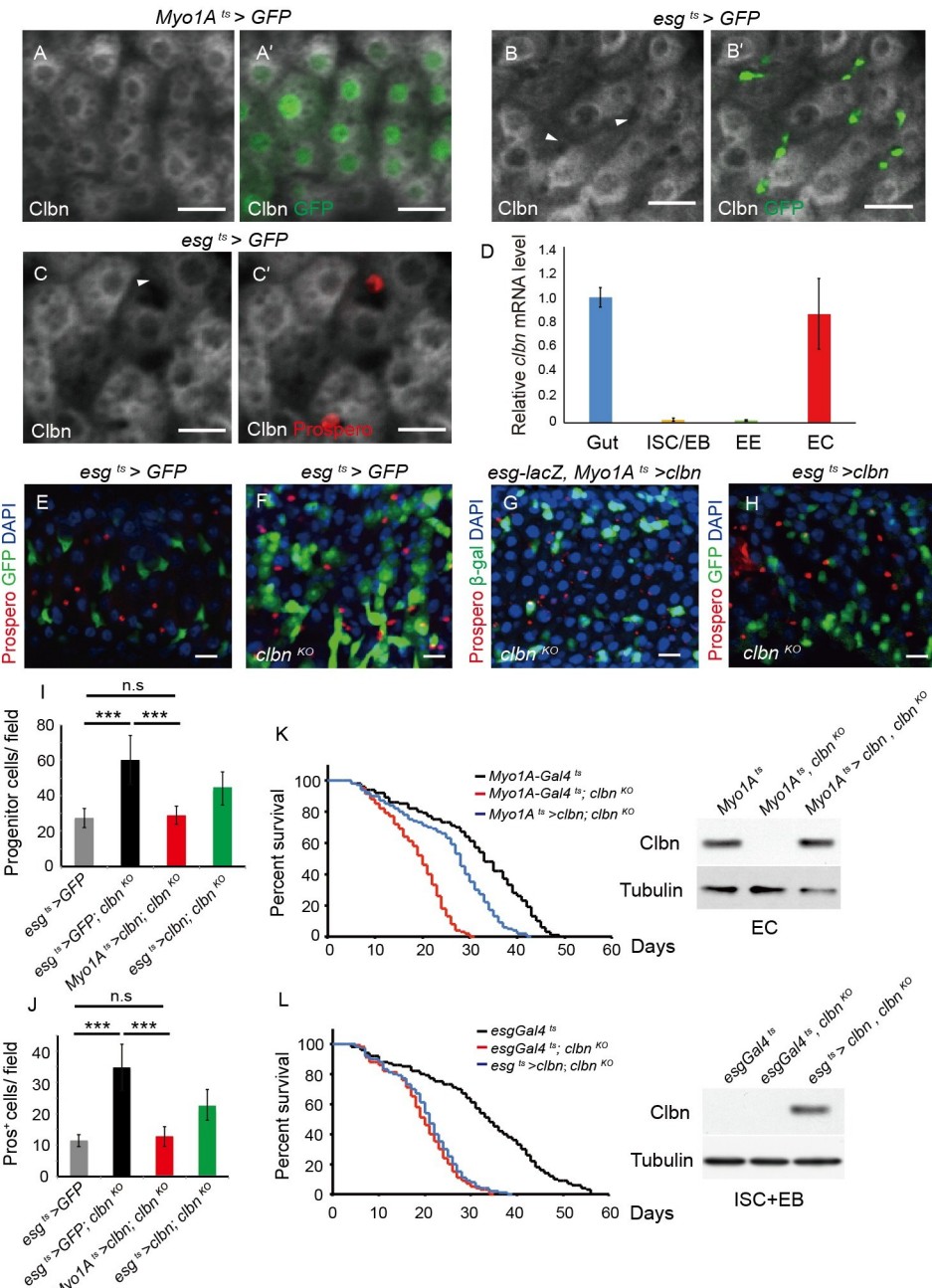

**Fig 2. Clbn is highly expressed in ECs and EC-specific expression of *clbn* rescues defect in *clbn* mutant.** (A-C')
Clbn localization in ECs detected with anti-Clbn antibody staining (white) in 5-day-old female midgut, EC cells were
marked with *Myo1A*<sup>ts</sup> > *GFP* (A', green), progenitor cells were marked with *esg*<sup>ts</sup> > *GFP* (B', green, arrowhead), and
EE cells were marked with anti-Prospero antibody staining (C', red, arrowhead). (D) Histogram of the relative mRNA
level of *clbn* in gut, ISC/EB, EE, and EC cells determined by RT-qPCR in three independent experiments
(mean ± SEM). (E-H) The posterior midguts of 15-day-old female control (*esg*<sup>ts</sup> > *GFP*) and *clbn*<sup>KO</sup> flies stained with
anti-Prospero antibody (red) and DAPI (blue). Expression of *clbn* in the posterior midgut of 15-day-old female flies
driven by *Myo1A*<sup>ts</sup> -*Gal4* (G) or *esg*<sup>ts</sup>-*Gal4* (H) under *clbn*<sup>KO</sup> background, and detected with anti-Prospero antibody
(red), anti-β-gal antibody (green) and DAPI (blue). (I) Quantification of the number of progenitor cells in control
(n = 12), *clbn*<sup>KO</sup> (n = 10), and flies expressing *clbn* driven by *Myo1A*<sup>ts</sup>-*Gal4* (n = 12) or *esg*<sup>ts</sup>-*Gal4* (n = 11) under *clbn*
<sup>KO</sup> background. (J) Quantification of the number of Prospero<sup>+</sup> cells in control (n = 12), *clbn*<sup>KO</sup> (n = 10), and flies
expressing *clbn* driven by *Myo1A*<sup>ts</sup>-*Gal4* (n = 12) or *esg*<sup>ts</sup>-*Gal4* (n = 11) under *clbn*<sup>KO</sup> background. The data shown are
means ± SEM, and *P* value was noted as follows: ***P* < 0.001. (K-L) Survival curves of indicated genotypes. The
expression of Clbn in indicated cell types was validated by Western-blot. Scale bars: 20 um.

experiments using intestinal extract, we observed highly enriched Clbn protein in the fraction containing mitochondria marker ATP5A (Fig 3D). To determine the sub-mitochondrial localization of Clbn, the mitochondria from midgut was purified and then digested with protease K. Clbn was completely removed by protease K treatment (Fig 3E), as that of Tom20, a marker of the outer membrane protein. In comparison, Cytochrome C and SOD2, which reside at the inter-membrane space and the matrix, respectively, were resistant to protease K treatment (Fig 3E). Thus, Clbn localizes to the mitochondrial outer membrane. Taken together, we concluded that Clbn resides at the outer membrane of mitochondria in ECs to perform its functions.

We next investigated the roles of Clbn in regulating mitochondrial morphology in ECs. In control flies, mitochondria form the net-like structure around the nucleus (Fig 4A), while in *clbn* mutant, mitochondria were highly fragmented (Fig 4B). In young flies (5-day-old), loss of *clbn* did not significantly affect the mitochondrial morphology (S5A Fig). However, *clbn* mutants at day 15 after birth exhibited fragmented mitochondria morphology (S5B Fig). Moreover, we analyzed the ultrastructure of mitochondria in ECs by TEM and found that loss of *clbn* led to fragmented mitochondria (Fig 4G and 4H), indicating that loss of *clbn* in flies change the mitochondrial dynamics in ECs. To determine whether Clbn generally regulate mitochondrial morphology in other tissues, we examined the mitochondrial morphology in flight muscle of the *clbn* mutant. Interestingly, no obvious changes of mitochondrial morphology were detected in flight muscle of *clbn* mutant when compared with the control (S5C and S5D Fig), indicating the function of Clbn in regulating the mitochondrial dynamics is EC-specific. To further confirm the role that Clbn plays for the maintenance of mitochondria is EC-specific, we performed rescue experiments by over-expressing *clbn* in ECs. Restoration of *clbn* expression in ECs successfully rescued the abnormal mitochondria morphology observed in *clbn* mutant (Fig 3C–3C"). Taken together, *clbn* is essential for the maintenance of mitochondria structure specifically in ECs.

Mitochondrial morphology is regulated by continuous fusion and fission. These dynamic processes are controlled by a group of GTPases conserved from yeast to human. Mitofusins play important roles in mediating fusion of outer mitochondrial membrane. The *Drosophila* mitofusin, also named Marf (mitochondrial assembly regulatory factor), plays a more general role to regulate mitochondrial fusion. While dynamin-related protein 1 (Drp1) is a major player responsible for mitochondrial fission [37–38]. To define whether Clbn and Drp1/Mfn work in the same pathway, we evaluated mitochondrial morphology in ECs upon knock-down of *drp1* or over-expression of *marf* in combination with *clbn* knock-out. Both *drp1* knock-down and *marf* over-expression in ECs can largely rescue mitochondrial fragmentation in *clbn* mutant (Fig 4C–4F, 4I and 4J). Knock-down of *drp1* can also partially rescue the ISC overproliferation in *clbn* mutant (S6 Fig). Furthermore, knock-down of *drp1* or over-expression of *marf* partially rescued the shortened lifespan of the *clbn* mutant (Fig 4K), indicating Clbn functions upstream of Drp1 and Mfn in regulating mitochondrial dynamics in ECs.

Clbn is the fly homolog of yeast RQC2 or Tae2. RQC2/Tae2 is a subunit of the ribosome-associated quality control (RQC) complex, which recruits the alanine- and threonine-charged tRNA to stalled ribosomes and is required for CAT- tailing in yeast [39]. In *Drosophila*, Clbn can also bind to Ala- and Thr-tRNAs, and loss of *clbn* rescued mitochondrial morphology, ATP production, and DA neuron loss phenotypes of *PINK1* mutant [35]. To examine whether the mitochondrial defects in *clbn* mutant is associated with the quality control of ribosomes, we over-expressed *dNOT4*, *dUPF1*, *deRF1* and *dXRN1* in *clbn* mutant, these RQC pathway genes work in the same pathway as PINK1 to control mitochondrial function in ECs [40]. Over-expression of these RQC factors in ECs had no effect on ISCs over-proliferation and mitochondrial fragmentation phenotypes of *clbn* mutant (S7 Fig and Fig 4L–4Q), indicating that defect in intestinal homeostasis in *clbn* mutant is independent of quality control of ribosomes.

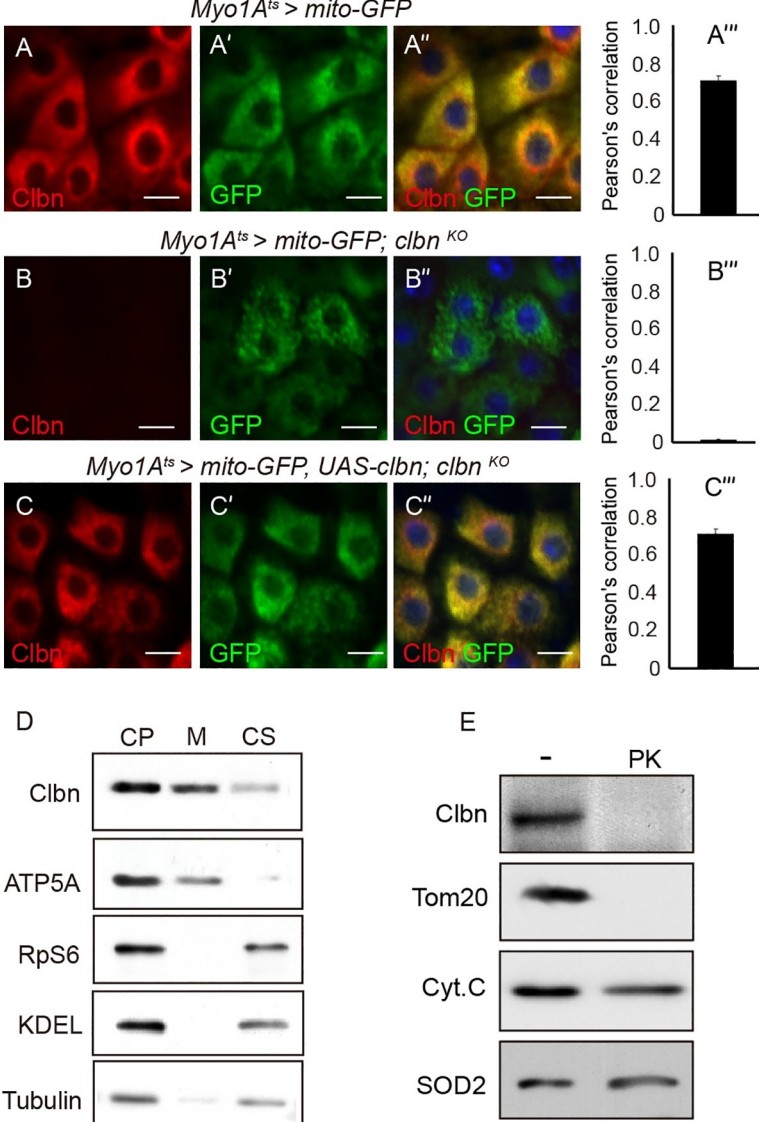

**Fig 3. Clbn is localized to mitochondria in ECs.** Mitochondria in ECs of 15-day-old female control (A-A") and *clbn*^*KO*^ flies (B-B") were labeled with mito-GFP (green) and stained with anti-Clbn antibody (red) and DAPI (blue). (C-C") Induced expression of *clbn* in *clbn*^*KO*^ ECs of 15-day-old females driven by *Myo1A*^*ts*^-*Gal4* and stained with anti-Clbn antibody (red) and DAPI (blue). Mitochondria were labeled with mito-GFP (green). Scale bars: 10 um. (A"'-C"') The degree of colocalization of Clbn and mitochondria was quantified by Pearson's correlation. (D) Western blot analysis of Clbn distribution in cytoplasmic (CP), mitochondrial (M) and cytosolic (CS) fractions. Samples were probed with anti-Clbn, anti-ATP5A, anti-RpS6, anti-KDEL, and anti-tubulin antibodies. RpS6 and KDEL were used as markers of the ribosome and endoplasmic reticulum (ER), respectively. (E) Submitochondrial localization of Clbn. Mitochondria from midguts were digested with or without protease K (PK). Protein extract was subjected to Western blot analysis. Tom20, Cytochrome C (Cyt.C), and SOD2 were used as markers of the outer membrane, inter-membrane space, and matrix, respectively.

## Multiple genes related with mitochondria are differentially expressed in *clbn* mutant

To investigate the intrinsic mechanisms of Clbn functions, we systemically analyzed genes expression in *clbn* mutant using a microarray assay. A whole genome transcripts with 18,952

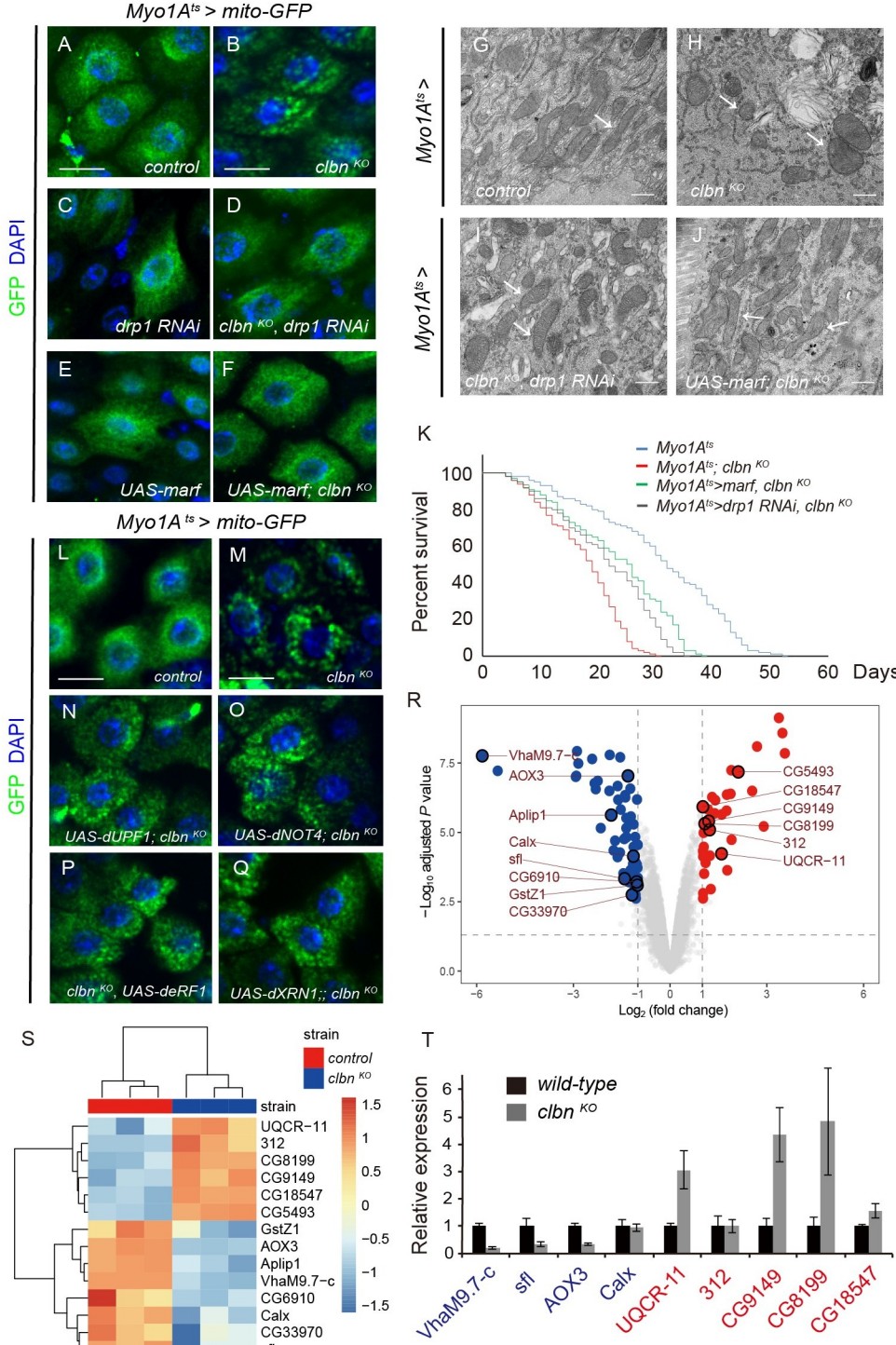

**Fig 4. Clbn regulates mitochondrial dynamics in ECs.** Mitochondria in ECs of 15-day-old female control (A) and *clbn*[KO] flies (B) were labeled with mito-GFP (green) and stained with DAPI (blue). (C-D) Knockdown of *drp1* in ECs of 15-day-old females driven by *Myo1A*[ts]-*Gal4* and stained with DAPI (blue). Mitochondria were labeled with mito-GFP (green). (E-F) Over-expression of *marf* in ECs of 15-day-old females driven by *Myo1A*[ts]-*Gal4* and stained with DAPI (blue). Mitochondria were labeled with mito-GFP (green). Scale bars: 10 um. (G-J) Representative electron microscopy images of ECs from 15-day-old female adults with indicated genotypes, arrow indicated mitochondria. Scale bar: 500 nm. (K) Survival curves of indicated genotypes. (L-Q) Mitochondria in ECs of control (I), *clbn*[KO] (J), over-expression of *dNOT4* (K), *dUPF1* (L), *deRF1* (M) and *dXRN1* (N) under *clbn*[KO] background were labeled with

mito-GFP (green) and stained with DAPI (blue). Scale bars: 15 um. (R) Volcano plot based on fold change and adjust *P* value of all transcripts in *clbn*<sup>KO</sup> flies. Genes expressed at similar level between wild-type and *clbn*<sup>KO</sup> flies were shown as grey dots. Genes with fold change $\geq$ 2 and adjusted $P < 0.05$ in *clbn*<sup>KO</sup> flies were depicted in blue dots (61 genes, down-regulated) and red dots (35 genes, up-regulated). Symbols of differential genes related with mitochondria were shown. (S) Heat map representation of differentially expressed genes related with mitochondria. The color key represents normalized gene expression. (T) RT-qPCR validation of differentially expressed genes from 15-day-old females guts. The down- and up-regulated genes identified by microarray analysis were shown in blue or red, respectively. The data shown are means ± SEM.

probe sets representing 12,874 genes were evaluated, which is more than 72.4% of fly genes (17,772 genes, dmel_r6.25_FB2018_06 from flybase). Among them, 35 genes were up-regulated and 61 genes were down-regulated in *clbn* mutant (S1 Table) using the parameters of adjusted $P < 0.05$ and fold-change $\geq$ 2. To identify the functional differences that reflect the transcriptional difference between the *clbn* mutant and wild-type fly, a gene set enrichment analysis (GSEA) with GO biological process was performed, gene sets with $P < 0.01$ were chosen to indicate significance. The GO biological processes correlated with top 15 up-regulated gene sets (n = 112) were associated with stem cell differentiation, DNA repair, RNA processing, ribosome biogenesis etc, while top 15 down-regulated genes sets (n = 150) were associated with monocyte chemotaxis, amino acid transportation, muscle contraction and metabolisms etc (S8 Fig, S2 Table).

Interestingly, multiple differentially expressed genes in *clbn* mutant have been reported or inferred with functions in mitochondria (Fig 4R and 4S), although the top 15 up-regulated and down-regulated GO pathways do not include mitochondria (S8 Fig). Among these genes, *sfl* is inferred to regulate mitochondrial organization [41], Aplip1 regulates axonal transport of mitochondria and JNK cascade [42], and 312, also named MALSU1, is inferred as a component of mitochondrial assembly of ribosomal large subunit 1 [41]. Some genes regulate metabolic functions related with mitochondria, i.e. UQCR-11 and CG8199 are inferred as a component of ubiquinol-cytochrome c reductase complex or mitochondrial alpha-ketoglutarate dehydrogenase complex respectively [41], and CG9149 is inferred with mitochondria with unknown functions [41]. Moreover, VhaM9.7-c and CG33970 have ATPase activity [41], Calx is a Na/Ca-exchange protein and regulates calcium ion binding [43], AOX3 [44], CG18547, CG5493, and CG6910 are inferred to regulate oxidation-reduction process and electron transport [41, 45] (Fig 4R and 4S). We further validated a number of these genes using RT-qPCR analysis. Indeed, most of genes we detected showed similar relative expression pattern as in microarray data (Fig 4T).

## Loss of *clbn* leads to mitochondrial defect

The functional output of mitochondria can be monitored by the mitochondrial membrane potential and ATP production. We used 5,5',6,6'-tetrachloro-1,1',3,3'- tetraethylbenzimidazolyl-carbocyanine iodide (JC-1), a more mitochondria specific cationic fluorescent dye, to quantify the mitochondrial membrane potential. At low membrane potential, JC-1 forms the green-fluorescent monomer, while at higher potential, JC-1 forms aggregate and shows red-fluorescence. We observed reduced JC-1 aggregate in *clbn* mutant (Fig 5A–5A"), indicating a mitochondrial depolarization in *clbn* mutant. Furthermore, the ATP level was significantly reduced in *clbn* mutant, which can also be rescued by restoration of *clbn* expression in ECs (Fig 5B).

Mitochondrial dynamics are closely connected with the level of ROS, and mitochondrial dysfunction could induce the production of ROS and activation of oxidative stress. We next monitored the ROS level in ECs using the ROS reporter *gstD1-GFP* and dihydroethidium

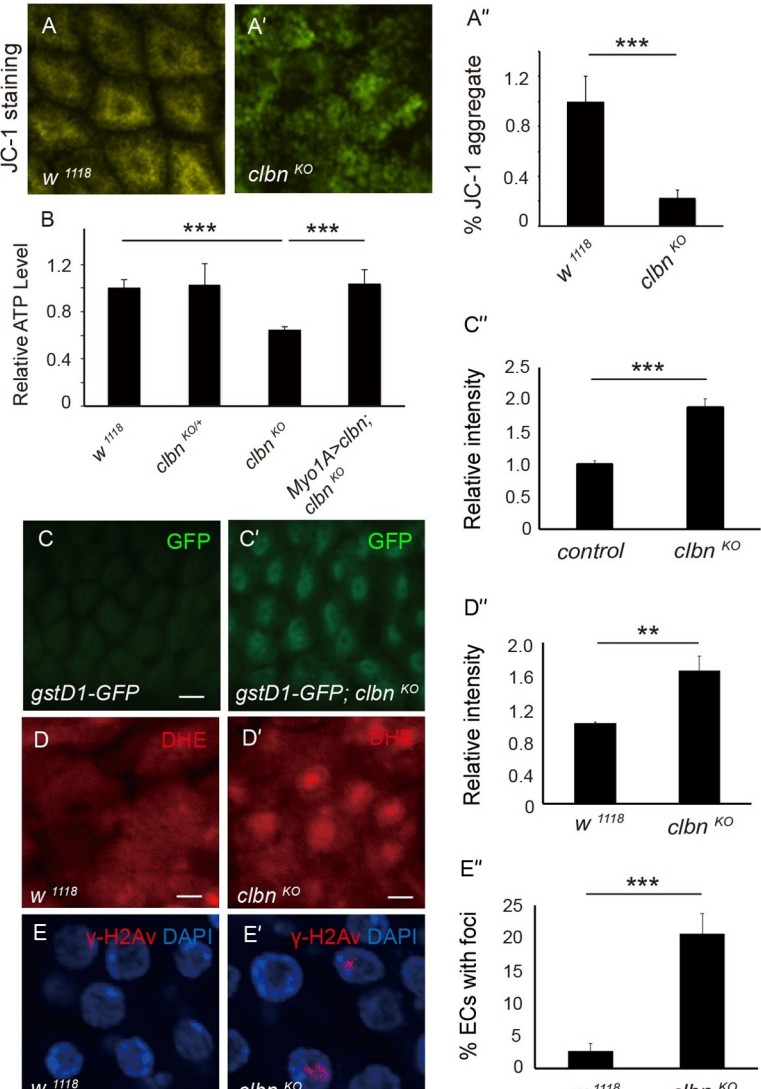

**Fig 5. Loss of *clbn* leads to mitochondrial defect.** (A-A') JC-1 staining of mitochondria in midgut in *wild-type* and *clbn*[KO] flies. (A") Quantification of the relative amount of JC-1 aggregate in control (n = 10) and *clbn*[KO] (n = 10) flies. (B) Quantification of the relative ATP level of guts with indicated genotypes. (C-C') *gstD1-GFP* expression in midguts of control and *clbn*[KO] flies. (C") Quantification of the fluorescence intensity. (D-D') Dihydroethidium (DHE) staining of midguts of *wild-type* and *clbn*[KO] flies. (D") Quantification of the fluorescence intensity (n = 10). (E-E') Midguts of *wild-type* and *clbn*[KO] flies were stained with anti-phosphorylated H2Av antibody (γ-H2Av) (red) and DAPI (blue). (E") Quantification of the γ-H2Av foci (n = 10). The data shown are means ± SEM, and *P* value was noted as follows: **$P < 0.01$, ***$P < 0.001$. All flies used in these essays were 15-day-old female adults.

(DHE) probe. As expected, increased expression of *gstD1-GFP* and DHE fluorescent intensity were observed in *clbn* mutant (Fig 5C–5D"). The accumulation of ROS in *clbn* mutant can be partially rescued by knock-down of *drp1* (S9 Fig), which is consistent with that knock-down of *drp1* partially rescued the ISC over-proliferation in *clbn* mutant (S6 Fig). Furthermore, we observed more γ-H2Av foci in ECs in *clbn* mutant (Fig 5E–5E"), the equivalent of mammalian γ-H2AX, indicating increased DNA damage in *clbn* mutant. All these data indicated that Clbn is an essential component of mitochondria in ECs and required for normal mitochondrial morphology and functions.

Increasing evidence suggests that the gut microbiota plays a vital role in mediating intestinal homeostasis. We also observed a high bacterial load in *clbn* mutant (S10A Fig), which is consistent with the loss of intestinal homeostasis in *clbn* mutant. To address the possibility that microbiota change contributes to the loss of intestinal homeostasis in *clbn* mutant, we reared germ-free *clbn* mutant and examined gut phenotype. The germ-free *clbn* mutant exhibited similar ISCs over-proliferation, mitochondrial fragmentation and ATP production phenotypes as the conventionally raised flies (S10B–S10H Fig), which confirmed that disrupted intestinal homeostasis in *clbn* mutant was caused by loss of *clbn* gene.

## Activation of JNK and JAK/STAT signaling pathways contributes to over-proliferation of ISCs in *clbn* mutant

We further investigated the cellular signaling pathways that are responsible for over-proliferation of ISCs in *clbn* mutant. As reactive oxygen species activate the Jun N-terminal kinase (JNK) signaling pathway in *Drosophila* [27, 28], we monitored JNK activity by using an enhancer-trap line, *puc-lacZ*, as a reporter and examined its expression in ECs. In control flies, *puc-LacZ* expression was low (Fig 6A). However, a significant increased expression of *puc-lacZ* was observed in *clbn* mutant (Fig 6A'). Moreover, the expression level of *puc-LacZ* in *clbn* MARCM clones was greatly increased when compared with that in surrounding control ECs (Fig 6B–6B').

In *Drosophila*, JAK-STAT, EGFR, WNT and Dpp/TGF-β signaling pathways play major roles to regulate ISCs proliferation in midgut, at physiological condition or in response to stress [36, 46, 47]. We therefore performed RT-qPCR and examined expression level of ligands for these pathways. We observed the ~ 8-fold increase of *unpaired 2* (*upd2*) expression, and ~ 19-fold increase of *unpaired 3* (*upd3)* expression, the two ligands of the JAK/STAT pathway, whereas expression level of *keren* (*krn*, a ligand for EGFR) and *dpp* (a ligand for TGF-β) was not significantly changed (Fig 6C). Consistently, expression of *upd-lacZ* reporter was greatly increased in *clbn* mutant clones (Fig 6D–6D'), while staining signal of anti-diphosopho-ERK antibody (dpERK), a direct readout of MAPK signaling activity, was not changed in *clbn* mutant (S11 Fig).

We also measured activity of JAK-STAT, WNT and Notch signaling pathways using reporter lines. Compared with control flies, expression of *10 × stat-GFP*, a Stat92E reporter, was greatly increased in the *clbn* mutant (Fig 6E–6E'), indicating that flies loss of *clbn* had aberrant activation of JAK-STAT signaling pathway. Moreover, knock-down of *stat92E* in ISCs/EBs can rescue the ISC over-proliferation phenotype in *clbn* mutant (Fig 6F–6F'). We did not observe any significant change of Notch or Wnt signaling pathways in *clbn* mutant, using a Notch responsive element (NRE) reporter *NRE-GFP* and a wingless reporter *wg-lacz* (S12 Fig). To provide further evidence that Clbn regulates ISCs proliferation non-autonomously, the wild-type control clones or *clbn* mutant clones were induced during adulthood. In the control clones, ISCs and EBs were distributed evenly along the midgut with high level of membrane-associated Armadillo/β-catenin protein (Fig 6G), and in *clbn* mutant clones, the aberrantly increased number of progenitor cells were present in proximity to the ECs (Fig 6G').

## Intestinal epithelial junctions are disrupted in *clbn* mutant

As the *clbn* mutant displayed defects in intestinal barrier (Fig 1B–1E) and increased proliferation of ISCs, we therefore assessed whether intestinal epithelial junctions were damaged in *clbn* mutant. Immunostaining with anti-Armadillo antibody to mark cell junctions revealed that Armadillo was strongly expressed around progenitor cells and presented at the cell membrane of ECs, and all cells in intestinal epithelium were well organized and assembled in the

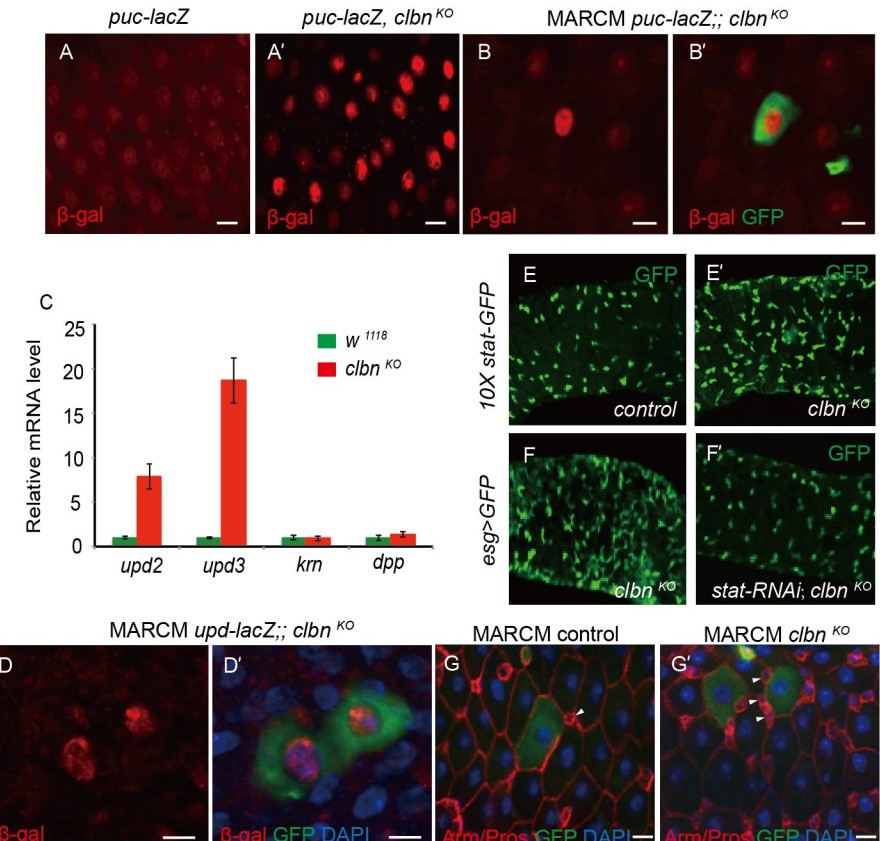

**Fig 6. Activation of JNK and JAK/STAT signaling pathways in *clbn* mutant midgut.** (A-A') Anti-β-gal antibody staining to show *puc-LacZ* in ECs of 15-day-old female control (A) and *clbn* $^{KO}$ (A') flies. (B-B') Adult midguts carrying MARCM clones of indicated genotypes were immunostained with anti-β-gal antibody (red). (C) Histogram of the relative mRNA level of *upd2*, *upd3*, *krn* and *dpp* in the guts of 15-day-old female control and *clbn* $^{KO}$ flies determined by RT-qPCR in three independent experiments (mean ± SEM). (D-D') Adult midguts carrying MARCM clones of indicated genotypes were immunostained with anti-β-gal antibody (red). (E-E') Expression of the $10 \times$ *stat-GFP* in 15-day-old female control and *clbn* $^{KO}$ flies. (F-F') The posterior midguts of 15-day-old female of indicated genotypes. Knock-down of *stat92E* in ISC/EB rescued the ISC over-proliferation phenotype in *clbn* mutant. (G-G') Adult midguts carrying MARCM clones of indicated genotypes were immunostained with anti-Armadillo antibody (red), anti-Prospero antibody (red) and DAPI (blue), arrowhead indicated progenitor cells. Scale bars: 15 um. Genotypes for MARCM: (B-B') *yw, hs-flp/+; UAS-GFP/+; tubGal4, FRT $^{82B}$, tubGal80/FRT $^{82B}$, puc-lacZ, clbn $^{KO}$*; (D-D') *yw, hs-flp/ upd-lacZ; UAS-GFP/+; tubGal4, FRT $^{82B}$, tubGal80/FRT $^{82B}$, clbn $^{KO}$*; (F-F') *yw, hs-flp/+; UAS-GFP/+; tubGal4, FRT $^{82B}$, tubGal80/FRT $^{82B}$, clbn $^{KO}$*.

control flies (Fig 7A and 7A'). However, in *clbn* mutant, the cellular architecture became aberrant and disorganized, and Armadillo protein presented at ECs membrane was decreased (Fig 7B and 7B'). The defect of cellular architecture in *clbn* mutant can be largely rescued through the EC-specific blocking of JNK activity by expression of a dominant negative form of Bsk (*UAS-bsk $^{DN}$*) (Fig 7C and 7C').

## Flies loss of *clbn* are susceptible to stress

Recent studies suggest that intestinal regeneration makes an important contribution to fly viability and life span [36]. Flies with intestinal dysplasia are short-lived, and animals with impaired ISCs proliferation or daughter cells differentiation die faster when challenged with pathogens, genotoxins, or ROS inducing compounds [48]. We therefore investigated

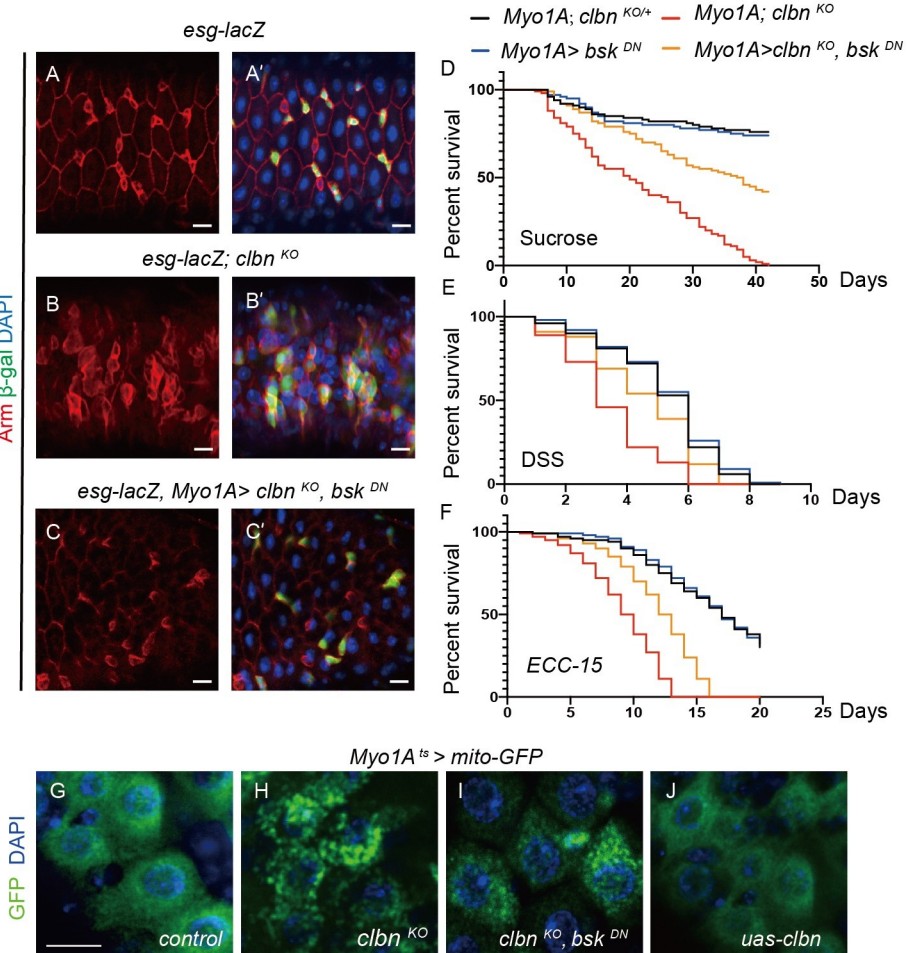

**Fig 7. *clbn* mutant displays disrupted intestinal integrity and increased mortality to stress.** Adult midguts of 15-day-old female control (A-A') and *clbn* KO (B-B') flies were immunostained with anti-Armadillo antibody (red) and DAPI (blue). (C-C') Blocking JNK activity in ECs using *bsk*DN largely rescued the defect in *clbn* KO flies. (D-F) Survival curves of flies of indicated genotypes fed with sucrose (D), DSS (E) or *Ecc15* (F). n > 100. Scale bars: 20 um. Mitochondria in ECs of 15-day-old female control (G), *clbn* KO (H), over-expression of *bsk* DN under *clbn* KO background (I) and over-expression of *clbn* (J) were labeled with mito-GFP (green) and stained with DAPI (blue). Scale bars: 20 um.

the ability of *clbn* mutant to resist stress. Under low nutrition condition, the *clbn* mutant had a significantly reduced lifespan: all *clbn* knock-out flies were dead at day-42 after eclosion, while more than 70% *clbn* heterozygous flies were still viable (Fig 7D). Moreover, the *clbn* mutant had greatly reduced viability when treated with dextran sodium sulfate (DSS) or enteropathogenic bacteria *ECC-15* (Fig 7E and 7F). Consistent with shortened lifespan, flies with loss of *clbn* had more progenitor cells in the midgut at low nutrient condition or when treated with DSS or *ECC-15* (S13 Fig). The increased mortality and over-proliferation of progenitor cells seen with low nutrition, DSS treatment or bacterial infection of *clbn* mutant can be largely rescued by inhibiting JNK signaling in ECs (Fig 7E and 7F and S13 Fig). On the other hand, blocking JNK signaling didn't rescue the defective mitochondrial morphology in ECs (Fig 7G–7I), indicating that mitochondrial fragmentation is an upstream event to Clbn depletion.

## Loss of *clbn* promotes tumor progression in gut

The sustained proliferative signaling and the reprogramming of energy metabolism are two of the hallmarks of cancer [49]. Our previous study demonstrated that over-expression of *clbn* in A549 lung cancer cell line can repress colony formation and invasiveness [32]. In addition, flies with loss of *clbn* have activated JAK-STAT signaling pathway which is responsible for cell proliferation (Fig 6), and changed mitochondrial metabolism (Fig 4). These findings promoted us to consider whether loss of *clbn* contributes to the tumor growth in response to tumorigenic stimuli in gut, although *clbn* knock-out flies generate no spontaneous tumors under physiological conditions. Over-activation of the mitogenic EGFR signaling pathway can establish a tumor model in the fly midgut [2, 50, 51], therefore, a constitutively active form of *Ras* oncogene at 85D (also known as $Ras^{V12}$) was induced by $esg^{ts}$>*Gal4* to activate EGFR signaling pathway in the progenitor cells. In the control flies, progenitor cells were randomly scattered along the basement membrane in the midgut (S14A–S14A' Fig). Consistent with aforementioned data, we observed robust proliferation of ISCs in *clbn* mutant (S14B Fig). The ISCs accumulated and attached to the basement membrane and had normal apical-basal polarity (S14B' Fig). Forced expression of $Ras^{V12}$ in progenitor cells for 3 days resulted in hyper-proliferation of ISCs and tumor formation, with accumulation of aberrantly differentiated progenitor cells and loss of epithelial cell polarity (S14C and S14C′ Fig). Depletion of *clbn* enhanced this effect, promoting tumor mass growth and invasion induced by $Ras^{V12}$ expression (S14D and S14E Fig). This suggests that dysfunctional mitochondria dynamics and metabolisms and disrupted intestinal homeostasis caused by loss of *clbn* might contribute to ISC tumor initiation or progression.

## Discussion

Modulation of mitochondrial dynamics and redox balance is vital for tissue homeostasis. Mitochondrial dynamics between the fusion and fission state play a critical role in determining mitochondrial morphology and functions. In this study, we established Clbn as an essential mitochondrial component in enterocytes that regulates mitochondrial dynamics, redox balance and intestinal homeostasis. Loss of *clbn* causes mitochondrial fragmentation, and blocking mitochondrial fission by knock-down of *drp1* or over-expression of *marf* can largely rescue the mitochondrial morphology in *clbn* mutant. These data suggest that Clbn promotes mitochondrial fusion and/or inhibits fusion by negatively regulating Drp1, and/or positively regulating Mfn. As the main source of intracellular ROS, mitochondrial damage could trigger ROS over-production and leads to oxidative stress and tissue damage [52]. Recent studies indicate that progenitor-specific mitochondrial dysfunction has significant impacts on ISCs proliferation, differentiation and tissue homeostasis in *Drosophila* [30, 31]. Loss of Nrf2 in ISCs causes accumulation of reactive oxygen species and deterioration of gut homeostasis [30]. Prevention of mitochondrial fusion by knock-down of *opa-1* or *marf* in intestinal progenitor cells can block differentiation into enterocytes [31]. Previous studies have shown that maintenance of mitochondria in ISCs is essential for tissue homeostasis [30, 31]; our data suggest that maintenance of mitochondrial dynamics and redox balance in ECs is also critical for ISCs proliferation and intestinal homeostasis. Mitochondrial dysfunction in ECs in *clbn* mutant results in lower mitochondrial membrane potential, less ATP production and higher level of ROS, which leads to ISCs over-proliferation and loss of intestinal homeostasis (Fig 8).

Our previous works have shown that Clbn has diversified functions, including nuclear exporting [32], DNA damage induced apoptosis [33] and cell cycle checkpoint [34]. A recent study also demonstrated that Clbn can bind to fly Ala- and Thr-tRNAs and involved in *PINK1* associated RQC pathway [35]. In this report, we identify a novel function of *clbn* in

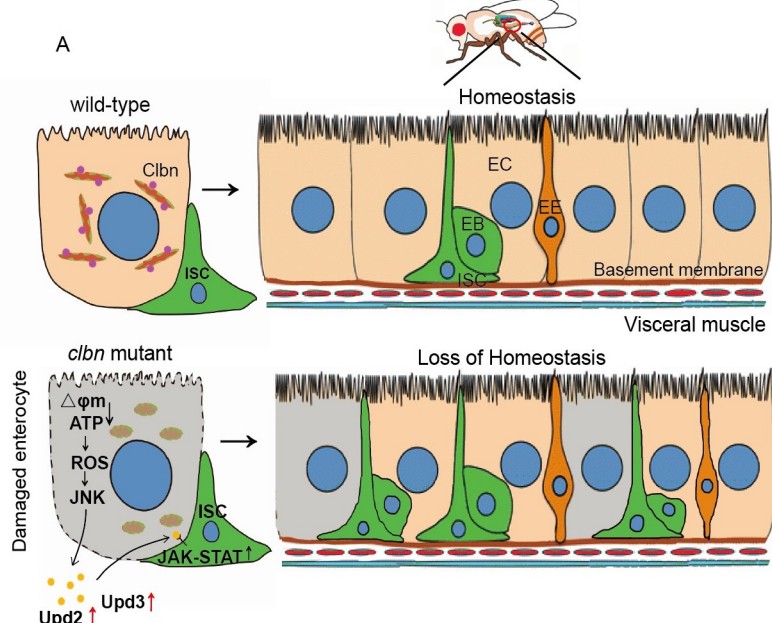

**Fig 8. Model of Clbn-dependent regulatory mechanisms in maintaining mitochondria dynamics and intestinal homeostasis.** Clbn mediates mitochondrial dynamics in ECs and removal of *clbn* leads to mitochondrial fragmentation and loss of intestinal homeostasis.

maintaining intestinal homeostasis by regulating mitochondrial dynamics in ECs. This function seems not to be general, but EC-specific, because we did not observe abnormal mitochondria in other tissues, such as flight muscle, from *clbn* knock-out flies. This idea is further supported by the data that EC-specific restoration of *clbn* expression can largely rescue the defect of gut and shortened lifespan of *clbn* mutant. The mitochondrial and gut defect observed in the *clbn* mutant should be a direct consequence of abnormal mitochondrial dynamics in ECs caused by *clbn* depletion. Several lines of evidence support this notion. Firstly, Clbn is localized to mitochondrial outer membrane in ECs, indicating a role in mitochondrial regulation. And blocking JNK activity in ECs partially rescued the gut defect and longevity of *clbn* mutant, but didn't rescue the defective mitochondrial morphology, indicating that mitochondrial fragmentation is an upstream event to Clbn depletion, but not a secondary effect caused by other defective processes. Secondly, ECs are terminally differentiated cells under cell cycle arrest, so the mitochondrial defect is less likely associated with the functions of Clbn in regulating G1/S transition. Lastly, over-expression of several factors in RQC pathways did not rescue mitochondrial morphology defect and ISCs hyper-proliferation in *clbn* mutant, which excludes the involvement of RQC in intestinal defects caused by *clbn* depletion.

Stem cell aging is a cause of organismal aging especially in high-turnover tissues such as intestine, and dysregulation of aging processes could promote tumor formation and progression. The incidence of spontaneous tumorigenesis in intestine is largely unaffected by mutation of *clbn* gene, however, the growth of intestine-tumor induced by activation of *Ras* gene was enhanced in *clbn* knock-out flies, indicating that increased oxidative stress induced by mitochondrial impairment could enhance *Ras*-driven epithelial tumorigenesis. Consistent with our results, a recent study demonstrated that blocking autophagy enhances $Ras^{V12}$-driven epithelial tissue over-growth via the accumulation of reactive oxygen species and activation of the JNK pathway [53]. Considering the versatility of Clbn functions, particularly its

importance in regulating redox balance and mitochondrial dynamics, the relative contribution of each of these processes in modulating stress response and tissue homeostasis remains to be determined. It is likely that Clbn could be a coordinating regulator that links these processes together in response to various stresses. It will be interesting to determine whether SDCCAG1, a homologue of Clbn, also regulates tissue homeostasis in high-turnover tissues in mammals. Our work may provide new insights in the mechanisms that control redox balance and tissue homeostasis.

## Materials and methods

### Fly genetics

All flies were maintained at 25°C on standard corn meal unless specified. Fly lines used in this study were as follows: *clbn $^{1Q}$ (clbn $^{KO}$)* and *UAS-clbn* were described in [32]; *esg-Gal4, UAS-GFP; Su(H)GBE-Gal4, UAS-GFP; Myo1A-Gal4; Gal80 $^{ts}$; delta-lacZ; upd-lacZ; wg-lacZ* and *10 × stat-GFP* were gifts from Dr. Lei Zhang, Shanghai Institute of Biochemistry and Cell Biology, Chinese Academy of Sciences; *pros-Gal4, mef2-Gal4, gstD1-GFP* and *mito-GFP* were gifts from Dr. Wanzhong Ge, Zhejiang University; *puc-lacZ* was gift from Dr. Renjie Jiao, Guangzhou Medical University; *act-Gal4, bsk $^{DN}$; NRE-GFP; UAS-Ras $^{V12}$; esg-lacZ; yw, hs-flp; UAS-GFP; tubGal4, FRT $^{82B}$, tubGal80; stat RNAi, drp1 RNAi, UAS-marf, UAS-dNOT4, UAS-dUPF1, UAS-deRF1, UAS-dXRN1* were obtained from Bloomington Stock Center.

### MARCM analysis

To generate MARCM clones during adulthood, progenies of the desired genotypes were collected within 3–5 days after eclosion and were heat-shocked in a 37°C water bath for two 60-min heat shocks over 2 days. The adult clones were examined 7–10 days later.

### Generation of anti-Pdm1 antibody

The cDNA fragment of Pdm1-aa 332–588 was amplified with primers pdm1-F: CCCAAGC TTGCCAACAGCCACGCAGTTCCA and pdm1-R: CCGCTCGAGTTTAGGGACTGTCC AGGGAGGGAT, and PCR product was constructed into Hind III/Xho I sites of pET-28a plasmid (Novagen), and protein expression was induced in BL21 competent cells. The gel slice corresponding to Pdm1 fusion protein was cut, crushed, emulsified with Freund's adjuvant and injected into rabbits to generate anti-Pdm1 antibody (Abgent Biotechnology, Suzhou, China). Sera were collected over a period of 2 months and purified with Protein A beads.

### Mitochondrial fractionation

The guts from 5-day-old female adults were dissected in cold PBS and cut into small pieces, then treated with 1× trypsin in EDTA (Thermo Scientific) for 2 h at room temperature and homogenized. Cytosolic and mitochondrial fractions were isolated following the manufacturer's instructions (Mitochondria Isolation Kit for Tissue, 89801, Thermo Scientific) and analyzed by Western blotting. The submitochondrial localization assay was performed as previously described [54]. The intact mitochondria were resuspended in 20 mM HEPES-KOH, pH 7.4, 0.6 M sorbitol. Protease K (100 μg/ml) was added to mitochondria preparation and kept on ice for 20 min. The reaction was stopped by the addition of 2 mM phenylmethylsulfonyl fluoride. Mitochondria were collected by centrifugation at 16,000 *g* for 5 min at 4°C and followed by SDS–PAGE and Western blot analyses.

## Immunostaining and imaging

The intestines from female adults were dissected in cold PBS and fixed immediately in PBS containing 4% (weight/vol) paraformaldehyde. Samples were then washed in PBS with 0.1% Triton X-100 (vol/vol) three times, blocked in PBTB [PBT containing 5% (vol/vol) normal goat serum], and incubated with primary antibodies overnight. The following primary antibodies were used: rabbit anti-Clbn (1:400), mouse anti-Prospero (1:200, Developmental Studies Hybridoma Bank), rabbit anti-Pdm1 (1:400), mouse anti-Armadillo (1:50, Developmental Studies Hybridoma Bank), rabbit anti-β-galactosidase (1:200, Promega), rabbit anti-phospho–histone-H3 (1:200, Millipore), rabbit anti-H2AvD pS137 (1:200, Rockland), rabbit anti-dpERK (1:200, Cell Signaling Technology), Alexa-Fluor-555-conjugated Phalloidin (1:500, Thermo Fisher Scientific). After three washes with PBT, secondary anti-rabbit or anti-mouse fluorescence antibodies, including Alexa 488 and 555 (1:400, Cell Signaling Technology), were used. Samples were mounted and analyzed on a Olympus FV1000 confocal laser-scanning microscope. The images were processed using Adobe Photoshop, Illustrator, and ImageJ.

## Transmission electron microscopy

The intestines from 15-day-old female adults were dissected in cold PBS and fixed immediately with 2.5% (vol/vol) glutaraldehyde with Phosphate Buffer (PB) (0.1 M, pH 7.4), washed four times in PB. Then tissues were immersed in 1% (weight/vol) $OsO_4$ and 1.5% (weight/vol) potassium ferricyanide aqueous solution at 4°C for 2 h. After washing with PB, tissues were dehydrated through graded alcohol (30, 50, 70, 80, 90, 100, 100%, 10min each) into pure acetone (2 × 10 min). Samples were infiltrated in graded mixtures (3:1, 1:1, 1:3) of acetone and SPI-PON812 resin (19.6 ml SPI-PON812, 6.6ml DDSA and 13.8ml NMA), then changed pure resin. Finally, tissues were embedded in pure resin with 1.5% BDMA and polymerized for 12 h at 45°C, 48 h at 60°C. The ultrathin sections (70 nm thick) were sectioned with microtome (Leica EM UC6), stained by lead citrate, and examined by a transmission electron microscope (FEI Tecnai Spirit120kV).

## JC-1 assay

Whole guts from 15-day-old female adults were dissected and transferred into 5 μM JC-1 (Thermo Scientific) in DMSO and incubated in dark for 30 min at room temperature. Samples were washed 2 times and mounted in PBS. Images were taken in the 568 nm channel using the Olympus FV1000 confocal laser-scanning microscope and the number of JC-1 red fluorescent aggregates was counted using ImageJ software.

## ATP measurement

For each measurement, 10 guts from 15-day-old female adults were dissected and the ATP level was measured using a luciferase-based bioluminescence assay (ATP Bioluminescence Assay Kit HS II; Roche Applied Science) according to manufacturer's instructions.

## DHE assay

Dihydroethidium (DHE) staining was performed as previously described [28]. Briefly, guts from 15-day-old female adults were dissected and handled in Schneider's medium throughout whole procedure. After incubation in 30 uM DHE (Invitrogen) for 5 min in dark at room temperature, samples were washed three times and mounted in Schneider's medium. Images were captured immediately with a Olympus FV1000 confocal laser-scanning microscope.

## Western blot

Protein extracts from mitochondrial fractionation were subjected to SDS/PAGE electrophoresis and transferred to polyvinylidene fluoride membrane. Membranes were immunoblotted with rabbit anti-Clbn (1:2,000), mouse anti-ATP5A (1:1,000, ab14748, Abcam), anti-KDEL (1:1,000, ab69659, Abcam), α-RpS6 (1:300, CST), α-Tom20 (1:800, #13929, CST), α-SOD2 (1:800, NB100-1992, Novus Biologicals), α-Cytochrome c (1:1,000, 7H8.2C12, Novus Biologicals), and mouse anti-tubulin (1:1,000, ab44928, Abcam) antibodies separately. Blots were subsequently incubated with horseradish peroxidase-conjugated goat-anti-rabbit or goat-anti-mouse secondary antibodies (GE Healthcare), and processed for chemiluminescence (GE Healthcare).

## FACS

FACS were performed as described previously [55]. Briefly, 200 guts were dissected in RNase-free PBS and dissociated for 45 min using elastase. Cells were then sorted using FACS Aria III (BD Biosciences) and sorted straight into RNA extraction buffer. The following fly lines were used to select different cell populations: *esg-Gal4 UAS-GFP; tub-Gal80ts* (ISCs and EBs), *tub-Gal80ts UAS-GFP; Pros-Gal4* (EEs), *Myo1A-Gal4; tub-Gal80ts UAS-GFP* (ECs).

## RT-qPCR

Total RNA was extracted from 10 guts of 15-day-old female adults using TRIzol (Invitrogen), and cleaned with RNAeasy kit (QIAGEN). The complementary DNA (cDNA) was synthesized using the iScript cDNA synthesis kit (Bio-Rad). Quantitative PCR was performed using the iScript one step RT-PCR SYBR green kit (Bio-Rad) for three independent biological replicates. The ribosomal gene *rp49* was used as the normalization control. All results were presented as mean ± SEM. Primers used for RT-qPCR were in S3 Table.

## Microarray assay and analysis

Embryos of *Oregon R* and *clbn* knock-out flies at 18 hr were collected and aged 2–4 h, and mRNA were isolated using Qiagen Oligotex Direct mRNA isolation kit. The microarray assay was performed with a Affymetrix CHP Analysis, and gene expression was evaluated using Genechip Drosophila Genome 2.0 Array (Affymetrix), which includes 18,952 probe sets representing 12,874 genes (R package drosophila2.db). Gene expression data was preprocessed using rma (R package affy). Differential expression analysis was performed using limma (R package limma) to identify individual genes demonstrating enrichment. Differentially expressed genes with adjusted $P < 0.05$ and fold-change $\geq 2$ were considered statistically significant. Volcano plot was used to visualize differential expression distribution. Functional enrichment analysis was performed to determine differences in gene sets between phenotypes (GSEA) using gsea function in R package clusterProfiler [56] with Gene Ontology biological process in msigdbr package. One thousand gene set permutations were run for each pathway. Minimal size of each gene set for analyzing is 10, and maximal size of genes annotated for testing is 500. Gene sets with $P < 0.01$ were chosen to indicate significance. Top 15 gene sets up-regulated and down-regulated were shown in bubble plot. Expression pattern of differentially expressed genes related with mitochondria, ATP, electron transport was shown in heat map.

## Smurf assay

Intestinal integrity was determined using the Smurf assay as described [57]. Briefly, female flies of each genotype were transferred onto fresh medium containing FD&C blue dye #1

(2.5% w/v) for 8 h. A fly was counted as a Smurf when dye coloration was observed outside of the digestive tract.

## Bacterial infection and DSS feeding assays

*Erwinia carotovora ssp. carotovora* 15 (*Ecc*15) was cultured in LB broth at 29˚C for 16 h. Fifteen-day-old female adults were starved in empty vials for 2 h at 25˚C, then transferred to vials containing a piece of 2.5 cm × 3.75 cm chromatography paper. Chromatography paper was wetted with 2.5% sucrose solution (control), *Ecc*15 bacteria (final OD600 = 100) in 2.5% sucrose, or 5% Dextran sulfate sodium (DSS) in 2.5% sucrose solution as feeding medium. Flies were transferred to new vials with freshly prepared feeding medium every day. After 3 days, guts were dissected and examined.

## Lifespan assay

The control flies and *clbn* $^{KO}$ flies were collected at eclosion. Fifteen flies were reared per tube to avoid overcrowding, and flies were transferred to a new tube every 3 days. Survival flies were checked on a daily base until natural death. At least 12 tubes per genotype were scored, and more than 100 flies were scored per genotype. Survival curves were plotted using Excel.

## Bacterial load

Flies were surface sterilized with 70% ethanol and then dissected on ice in sterile PBS. Dissected intestines were stored in sterile eppendorf tubes at -80˚C prior to DNA extraction and were then extracted using the PowerSoil DNA isolation kit (MoBio). The q-PCR reactions were conducted in three independent biological replicates using SYBR Green PCR master mix (Takara). Universal primers for the 16 S ribosomal RNA gene were against variable regions 1 (V1F) and 2 (V2R). The *Drosophila* Actin 5C gene was used for normalization. PCR primers are shown in S1 Table.

## Generation of axenic flies

Germ-free flies were generated following the previous report with minor modification [58]. Three to 5 days old flies were transferred on fresh fruit juice agar plates. After 1 day of habituation, flies were allowed to lay eggs for 4–6 hours. Eggs were washed with 70% EtOH, sterile water and dechorionated using 10% bleach for ~ 10 min. Eggs were then transferred under a sterile flow hood and maintained on axenic food until adult stage.

## Statistics

All statistical comparisons were performed using data collected from three biological replicates with origin 9.0. Unpaired two-tailed Student's t-test was used to assess statistical significance between two genotypes/conditions and one-way ANOVA for comparison of multiple samples. The significance level was indicated as $^*$ for $P < 0.05$, $^{**}$ for $P < 0.01$, and $^{***}$ for $P < 0.001$.

## Supporting information

**S1 Fig. Loss of *clbn* causes increased number of progenitor cells and enteroendocine cells in flies older than 15 days.** The posterior midguts of 5-day-old, 15-day-old and 30-day-old female control (A-C) and *clbn* $^{KO}$ flies (D-F) stained with anti-Prospero antibody (red) and DAPI (blue). (G-H) Quantification of the number of progenitor cells (ISCs and EBs) (G) or EEs (H) in control (n = 10) and *clbn* $^{KO}$ flies (n = 10). (I-J) Quantification of the number of PH3$^+$ (I) or Prospero$^+$ cells (J) in the whole gut of control (n = 10) and *clbn* $^{KO}$ flies (n = 10).

The data shown are means ± SEM, and *P* value was noted as follows: $^*P < 0.05$, $^{***}P < 0.001$. Scale bars: 20 um.
(TIF)

**S2 Fig. Loss of *clbn* causes increased number of ISCs and EBs.** (A-C) The posterior midguts of 15-day-old female control (*delta-lacZ*), *clbn*$^{KO/+}$ and *clbn*$^{KO}$ flies were stained with anti-β-gal antibody. (D-F) The posterior midguts of 15-day-old female control (*Su(H)GBE >GFP*), *clbn*$^{KO/+}$ and *clbn*$^{KO}$ flies. (G-H) Quantification of the number of ISCs (G) or EBs (H) in control (n = 10), *clbn*$^{KO/+}$ (n = 10) and *clbn*$^{KO}$ flies (n = 10). The data shown are means ± SEM, and *P* value was noted as follows: $^{***}P < 0.001$. Scale bars: 20 um.
(TIF)

**S3 Fig. Clbn is dispensable for cell differentiation in midgut.** MARCM clones in control (A-A', C-C') or *clbn*$^{KO}$ flies (B-B', D-D') were immunostained with anti-Prospero antibody (A-B), anti-Pdm1 antibody (C-D) and DAPI. Clones were marked by GFP (green), EEs by Prospero (red), and ECs by Pdm1 (red). Scale bars: 15 um. (E-F) Quantification of the number of EEs (E) and ECs (F) in clones of control and *clbn* mutant. The EE/EC cell numbers were normalized with the average clone size. (G-G') MARCM clones in *clbn*$^{KO}$ flies were immunostained with anti-Clbn antibody and DAPI. Clones were marked by GFP (green). Scale bars: 15 um. (H) Quantification of number of cells in MARCM clones in control (n = 40) and *clbn*$^{KO}$ (n = 40) flies. Genotypes: (A-A', C-C') *yw, hs-flp/+; UAS-GFP/+; tubGal4, FRT*$^{82B}$*, tubGal80/ FRT*$^{82B}$; (B-B', D-D') *yw, hs-flp/+; UAS-GFP/+; tubGal4, FRT*$^{82B}$*, tubGal80/ FRT*$^{82B}$*, clbn*$^{KO}$.
(TIF)

**S4 Fig. EC-specific knock-down of *clbn* results in increased ISCs proliferation.** (A-E) The posterior midguts of flies of 15-day-old female control (*esg-lacz)* (A), *clbn* knock-down in ECs (B), progenitor cells (C), EEs (D), and visceral muscle (E) were stained with anti-β-gal antibody (green), anti-Prospero antibody (red) and DAPI (blue). (F) Quantification of the number of progenitor cells in flies of control (n = 12), ECs knock-down of *clbn* (n = 10), ISCs and EBs knock-down of *clbn* (n = 10), EEs knock-down of *clbn* (n = 10), and visceral muscle knock-down of *clbn* (n = 10). (G) Quantification of the number of Pros$^+$ cells in flies of control (n = 12), ECs knock-down of *clbn* (n = 10), ISCs and EBs knock-down of *clbn* (n = 10), EEs knock-down of *clbn* (n = 10), and visceral muscle knock-down of *clbn* (n = 10). The data shown are means ± SEM, and *P* value was noted as follows: $^{***}P < 0.001$. Scale bars: 20 um.
(TIF)

**S5 Fig. Clbn is localized to mitochondrial outer membrane in ECs and loss of *clbn* leads to mitochondrial fragmentation in ECs but not in flight muscle.** Mitochondria in the ECs of 5-day-old (A) and 15-day-old *clbn*$^{KO}$ female flies (B) were labeled with mito-GFP (green) and stained with DAPI (blue). Mitochondria in the flight muscle of 15-day-old female control (C) and *clbn*$^{KO}$ flies (D) were labeled with mito-GFP (green) and stained with Phalloidin (red) and DAPI (blue). Scale bars: 20 um.
(TIF)

**S6 Fig. Knockdown of *drp1* rescues ISC over-proliferation in clbn mutants.** (A-B) The posterior midguts of 15-day-old female flies of indicated genotypes were stained with anti-β-gal antibody (green). (C) Quantification of the number of *lacZ*$^+$ cells in flies of indicated genotypes (n = 10). The data shown are means ± SEM, and *P* value was noted as follows: $^{***}P < 0.001$.
(TIF)

**S7 Fig. Loss of *clbn* leads to ISCs over-proliferation and mitochondrial fragmentation independent of ribosome-associated quality control pathways.** (A-F) The posterior midguts of 15-day-old female flies of control (*esg-lacz*) (A), *clbn*$^{KO}$ (B), over-expression of *dNOT4* (C), *dUPF1* (D), *deRF1* (E) and *dXRN1* (F) in ECs under *clbn*$^{KO}$ background were stained with anti-β-gal antibody (green), anti-Prospero antibody (red) and DAPI (blue). (G) Quantification of the number of progenitor cells in flies of control (n = 10), *clbn*$^{KO}$ (n = 10), over-expression of *dNOT4* (n = 10), *dUPF1* (n = 10), *deRF1* (n = 10) and *dXRN1* (n = 10) in ECs under *clbn*$^{KO}$ background. (H) Quantification of the number of Pros$^{+}$ cells in flies of control (n = 10), *clbn*$^{KO}$ (n = 10), over-expression of *dNOT4* (n = 10), *dUPF1* (n = 10), *deRF1* (n = 10) and *dXRN1* (n = 10) in ECs under *clbn*$^{KO}$ background.
(TIF)

**S8 Fig. Bubble plot based on GSEA analysis shows top 15 gene sets up-regulated and down-regulated.** Biological processes were ranked as enrichment score of each gene sets. The color of bubbles represents enrichment score. The size of bubbles represents–log10 (*P* value).
(TIF)

**S9 Fig. Knockdown of *drp1* rescues ROS accumulation in *clbn* mutants.** (A-B) Dihydroethidium (DHE) staining of midguts of indicated genotypes. (C) Quantification of the DHE fluorescence intensity (n = 10). The data shown are means ± SEM, and *P* value was noted as follows: $^{**}P < 0.01$.
(TIF)

**S10 Fig. Microbiota does not contribute to the intestinal defects in *clbn*$^{KO}$ flies.** (A) Level of bacteria in guts of *wild-type* and *clbn*$^{KO}$ flies was detected by qPCR for bacterial 16S rDNA. (B-C) The posterior midguts of 15-day-old germ-free female control (B) and *clbn*$^{KO}$ flies (C) were stained with anti-Prospero antibody (red) and DAPI (blue). (D-E) Mitochondria in ECs of 15-day-old germ-free female control (D) and *clbn*$^{KO}$ flies (E) were labeled with mito-GFP (green) and stained with DAPI (blue). (F-G) Quantification of the number of ISCs and EBs (F) or EEs (G) in control (n = 10) and *clbn*$^{KO}$ flies (n = 10). (H) Quantification of the relative ATP level in guts of indicated genotypes. Scale bars: 20 um.
(TIF)

**S11 Fig. The epidermal growth factor receptor (EGFR) pathway is not induced in the *clbn* mutant.** The posterior midguts of 15-day-old female control (*esg>GFP*, A-A'), *clbn*$^{KO}$ (B-B') and Ras over-expression (C-C') flies were stained with antibody against the diphospho-form of the extracellular signal-regulated kinase (dpERK) (red). Scale bars: 15 um.
(TIF)

**S12 Fig. The Notch and Wnt signaling activity is not changed in the *clbn* mutant.** Expression of *NRE-GFP* (Notch activity reporter) and *wg-lacZ* (Wnt activity reporter) was detected in posterior midguts of 15-day-old female *control* (A, C) and *clbn*$^{KO}$ (B, D) flies. Scale bars: 20 um.
(TIF)

**S13 Fig. Loss of *clbn* results in more progenitor cells in response to various stress.** Representative images of posterior midguts of 15-day-old female flies of indicated genotypes, 72 h after feeding with sucrose (A-B, G-H), DSS (C-D, I-J) or *ECC-15* (E-F, K-L). The progenitor cells were labeled with *esg-lacZ* (green). Scale bars: 20 um. Quantification of the number of PH3$^{+}$ (M) and Prospero$^{+}$ (N) cells in the whole gut of indicated genotypes (n = 15).
(TIF)

**S14 Fig. Loss of *clbn* promotes ISCs tumor growth.** (A-D) Progenitor cells of midguts from 15-day-old female flies of indicated genotypes were labeled by GFP. (A'-D') Sagittal view of the midgut from flies of indicated genotypes. GFP driven by *esg^{ts}-Gal4* (green) marks the progenitor cells, Phalloidin (red) marks the visceral muscle and brush border, and DAPI (blue) highlights the nuclei. (E) Cells per ISC tumor induced by *Ras^{v12}* in control (n = 12) or *clbn^{KO}* (n = 13) flies. The data shown are means ± SEM, and *P* value was noted as follows: $^{**}P < 0.01$. (TIF)

**S1 Table. Up and down-regulated genes in *clbn* mutant.**
(XLSX)

**S2 Table. List of gene sets for GSEA analysis.**
(XLSX)

**S3 Table. Primers for RT-qPCR.**
(XLSX)

# Acknowledgments

We thank Drs. Wanzhong Ge, Renjie Jiao, Lei Zhang, the Bloomington Stock Center for fly stocks, Xixia Li and Xueke Tan for helping with electron microscopy sample preparation and taking TEM images at the Center for Biological Imaging (CBI), Institute of Biophysics, Chinese Academy of Sciences, Dr. Paul Badenhorst for microarray assistance, Dr. Guangchuang Yu and FigureYa (Xiao Ya Hua Tu) for the figure technology support, Cindy Lim and members of the Bi laboratory for advice and discussions.

# Author Contributions

**Data curation:** Zhaoxia Dai, Dong Li, Xiao Du.

**Formal analysis:** Zhaoxia Dai, Dong Li.

**Funding acquisition:** Dong Li, Xiaolin Bi.

**Project administration:** Xiaolin Bi.

**Software:** Ying Ge.

**Writing – original draft:** Dong Li.

**Writing – review & editing:** Deborah A. Hursh, Xiaolin Bi.

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
