## [Decision Letter · Decision Letter 0]

24 Mar 2020

Dear Dr Bi,

Thank you very much for submitting your Research Article entitled 'Drosophila Caliban preserves intestinal homeostasis and lifespan through regulating mitochondrial dynamics and redox state in enterocytes' to PLOS Genetics. Your manuscript was fully evaluated at the editorial level and by independent peer reviewers. The reviewers appreciated the attention to an important problem, but raised some substantial concerns about the current manuscript. Based on the reviews, we will not be able to accept this version of the manuscript, but we would be willing to review again a much-revised version. We cannot, of course, promise publication at that time.

Should you decide to revise the manuscript for further consideration here, your revisions should address the specific points made by each reviewer. In particular, more than one reviewer would like to see stronger evidence for mitochondrial localization of Clbn. We will also require a detailed list of your responses to the review comments and a description of the changes you have made in the manuscript.

If you decide to revise the manuscript for further consideration at PLOS Genetics, please aim to resubmit within the next 60 days, unless it will take extra time to address the concerns of the reviewers, in which case we would appreciate an expected resubmission date by email to plosgenetics@plos.org.

[LINK]

We are sorry that we cannot be more positive about your manuscript at this stage. Please do not hesitate to contact us if you have any concerns or questions.

Yours sincerely,

Bingwei Lu

Associate Editor

PLOS Genetics

Gregory P. Copenhaver

Editor-in-Chief

PLOS Genetics

Reviewer's Responses to Questions

**Comments to the Authors:**

Reviewer #1: In this paper, the authors identified fly Caliban (Clbn) as a specific regulator of mitochondrial dynamics in enterocytes (ECs) and a key modulator of intestinal homeostasis. Clbn inactivation leads to mitochondrial fragmentation, ROS accumulation, intestinal stem cell overproliferation and shortened life span. Overall, this is an interesting study with well-controlled results. However, before recommending publication the following points need be addressed:

1. If intestinal homeostasis defects in clbn mutants are mainly due to mitochondrial fragmentation in ECs, restoring mitochondrial dynamics should rescue the intestinal phenotypes. Indeed, the authors found that preventing mitochondrial fission through drp1 knockdown or marf overexpression can partially rescue the shortened lifespan phenotype of the clbn mutants. It remains unclear whether drp1 knockdown can rescue other important phenotypes, such as ROS accumulation and ISC overproliferation, in clbn mutants.

2. The authors propose that ISC overproliferation in clbn mutants are due to upregulation of JAK-STAT signaling, which is in turn promoted by increased release of Upd from clbn mutant ECs. It would be important to further confirm this notion by testing whether downregulation of JAK-STAT signaling specifically in ISCs (for example through esg-Gal4>UAS-stat-RNAi) can rescue the ISC overproliferation phenotype in clbn mutants.

3. If clbn loss lead to mitochondrial fission and fragmentation in ECs, can Clbn overexpression results in mitochondrial fusion?

4. Some experiments in the paper, such as lifespan assay shown in Fig.1 and survival assay shown in Fig.6, did not show N number in the figure legends or main text.

Reviewer #2: In this paper, Di et al investigate the functions of Caliban (Clbn), with a focus on functions in the adult gut. The gene is clearly important and the phenotypic analysis is quite good, with a number of interesting cellular phenotypes detailed in a competent way using up-to-date methods. They postulate that Clbn functions in mitochondria, and that it’s action in gut enterocytes is critical for viability and normal lifespan. The later point is well supported by a very nice gut-specific lifespan rescue result shown in Fig 2 shown. In addition, interesting, convincing results showing various stress responses are presented in Fig 4, 5, and 6. However, the data relating to a mitochondrial-specific function of Clbn (Fig 3, 4) is inconclusive and doesn’t sufficiently support the authors’ conclusions. This significantly detracts from the papers’ general significance as a gene-function study. In addition, the paper is not very well written, and has many shortcomings in grammar, organization, textual presentation. The figures are well done, however. Detailed comments follow:

1. The authors claim that clbn is localized in mitochondria because: 1) It has a peri-nuclear stain and co-localized with mito-GFP; 2) Clbn protein fractionated with the mitochondria marker ATP5A. However, a pervious study (Doghman-Bouguerra, M., & Lalli, E. BBA. 2019) showed this protein associates with ribosomes, probably on the ER. ER is known to interact with mitochondria, and mitochondria-associated membranes (MAM) are important for mitochondria dynamics. MFN2 and Drp1 are on the MAM. Therefore, Clbn might well be ER- or MAM-localized. The images shown here are not high resolution enough to tell.

2. It is interesting that Clbn does not affect mitochondria in flight muscle, this may be because there is no ER-mitochondria interaction in the muscle. If the authors study MAM, it will make this paper so much better and more impact. In Fig3 D,E, one can see evidence of lost MAM. The authors should consider doing experiments focused on whether Clbn functions in MAM.

3. In the Western blot of mitochondria fractions, a ribosome associated protein and an ER associated protein should be used as negative controls, to rule out localizations there.

4. If the author can do a immune-TEM for Clbn localization, this might resolve the issue of its true localization.

5. The authors state that “The regenerative potential of stem cells is closely correlated with the level of intracellular reactive oxygen species (ROS). Normal physiological levels of ROS are essential for stem cells functions and fate decision.” Citations supporting this notion are missing.

6. In Fig 3S (MARCM clones), the authors should measure the clone size or cell number. The mutation promotes ISC proliferation, so the clone size should be bigger and with more cells. Please normalize the EE/EC cell numbers with total cell number or clone size.

7. JC-1 staining should be included in Fig 4.

8. The introduction does not establish that Clbn is a mitochondrial protein, nor does it introduce mutant allele or summarize previous work using this allele.

9. The lifespan rescues in Fig 2 are on a very different time scale than the lifespans shown in figure Fig 1. How do the authors account for this difference? Please discuss.

10. The data shown in Fig 7 (tumor interaction) is not quantitative, and is therefore inconclusive.

Reviewer #3: uploaded as an attachment

**Have all data underlying the figures and results presented in the manuscript been provided?**

Reviewer #1: Yes

Reviewer #2: No: No raw data is presented for the graphs.

Reviewer #3: Yes

PLOS authors have the option to publish the peer review history of their article (what does this mean?). If published, this will include your full peer review and any attached files.

Reviewer #1: No

Reviewer #2: No

Reviewer #3: No

---

## [Decision Letter · Decision Letter 1]

3 Sep 2020

Dear Dr Bi,

We are pleased to inform you that your manuscript entitled "Drosophila Caliban preserves intestinal homeostasis and lifespan through regulating mitochondrial dynamics and redox state in enterocytes" has been editorially accepted for publication in PLOS Genetics. Congratulations!

Please note that both Reviewers #1 and #2 have some suggestions for re-organizing some of figures (moving items between main figures and supplementary material).  This is largely a stylistic issue so we will leave this up to your discretion, but we would like to point out the both reviewers recommend shifting Fig S5A and B into the main figures (in our experience if multiple reviewers recommend something, it is likely many of your readers would also find the change helpful).  Feel free to make these changes in the final draft that you prepare for the production team (the editorial team will not need to re-evaluate).

Yours sincerely,

Bingwei Lu

Associate Editor

PLOS Genetics

Gregory P. Copenhaver

Editor-in-Chief

PLOS Genetics

Comments from the reviewers (if applicable):

Reviewer's Responses to Questions

**Comments to the Authors:**

Reviewer #1: In the revised version, the authors have sufficiently addressed my concerns.

One minor point: Since the subcellular distribution of Clbn within ECs is very important, it would be ideal if new results in Fig S5A and S5B could are moved to main Fig 3.

Reviewer #2: In this interesting paper, Dai et al report a cell-type specific function for Caliban (Clbn) in enterocytes of the Drosophila gut. They present a number of interesting observations, including mitochondria fragmentation and ROS accumulation after Clbn loss, and they show that this mitochondrial could stress causes Upd induction, JAK-Stat pathway activation, and intestinal stem cell (ISC) proliferation. The fly lifespan is also negatively affected. Although the original data was all of good quality, the sub cellular localization of Clbn was not clear, and the reviewers had some question as to the specificity of the phenotypes to mitochondrial functions. This point is addressed somewhat in the revision by new data showing the Clbn can be detected on the mitochondrial outer membrane, though much is also present in the cytosol (Fig S5AB). Unfortunately, the mechanisms via which Clbn KO causes mitochondrial fragmentation and ROS generation are still unclear, but these issues are difficult to address and can be attacked in future research on this conserved protein. Several other follow up experiments addressing the reviewers’ comments are provided. Overall, the response to the reviews is conscientious. The revised paper is a complete story that will contribute to the general understanding of mitochondrial function, as well as gut homeostasis. We would like to recommend that it be published. A couple of specific comments:

1. I think the new data in Fig S5A and B should be added to a main figure. The most notable long-lasting discoveries in this paper will likely be those in cell biology, as opposed to fly midgut function, and so data pertaining to the molecular and cellular function of Clbn is most valuable.

2. Likewise, the data in Figs S7 (I-N) and S14 are also valuable and should be moved to main figures.

3. The data on tumor growth (Fig 7C-E) is not very important (in my opinion) and could be moved to the supplement.

4. I would like to see some better, higher magnification, higher resolution, 3D reconstructed pictures of the mitochondrial fragmentation phenotype. This should be straightforward using a modern confocal microscope. Some more, more highly magnified TEMs of this would also be a nice addition.

Reviewer #3: The manuscript has been considerably improved. The authors have addressed my concerns.

**Have all data underlying the figures and results presented in the manuscript been provided?**

Reviewer #1: Yes

Reviewer #2: Yes

Reviewer #3: Yes

PLOS authors have the option to publish the peer review history of their article (what does this mean?). If published, this will include your full peer review and any attached files.

Reviewer #1: **Yes: **Yan Song

Reviewer #2: **Yes: **Bruce A Edgar

Reviewer #3: No

**Data Deposition**

http://datadryad.org/submit?journalID=pgenetics&manu=PGENETICS-D-20-00223R1

**Press Queries**

---

## [Editor Report · Acceptance letter]

8 Oct 2020

PGENETICS-D-20-00223R1 

Drosophila Caliban preserves intestinal homeostasis and lifespan through regulating mitochondrial dynamics and redox state in enterocytes 

Dear Dr Bi, 

We are pleased to inform you that your manuscript entitled "Drosophila Caliban preserves intestinal homeostasis and lifespan through regulating mitochondrial dynamics and redox state in enterocytes" has been formally accepted for publication in PLOS Genetics! Your manuscript is now with our production department and you will be notified of the publication date in due course.

With kind regards,

Jason Norris

PLOS Genetics

On behalf of:
